# Comparison of compensatory shoulder movements, functionality and satisfaction in transradial amputees fitted with two prosthetic myoelectric hooks

**Amélie Touillet, Constance Billon-Grumillier👤\*, Jonathan Pierret👤, Pierrick Herbe, Noël Martinet, Isabelle Loiret, Jean Paysant**

Centre Louis Pierquin, Institut Régional de Médecine Physique et de Réadaptation, UGECAM, Nancy, Nord-Est, France

\* constance.billon@ugecam.assurance-maladie.fr

## Abstract

The functionalities of myoelectric hooks, such as whether they allow wrist movements, as well as the volume and design of the devices, may impact how fitted transradial amputees use their upper limbs. The aim of the current study was to compare two prosthetic myoelectric hooks in terms of compensatory shoulder movements, functionality and user satisfaction. This monocentric, randomized, controlled, cross-over trial evaluated eight transradial amputees fitted with two prosthetic myoelectric hooks, the Greifer and the Axon-Hook, during two consecutive periods. At the end of each period, shoulder abduction (mean and percentage of time with shoulder abduction > 60˚) and manual dexterity were assessed using the Box and Blocks Test (BBT) on both sides, and satisfaction was assessed with the Evaluation of Satisfaction with Assistive Technology questionnaire. For each patient, data obtained with the BBT on the amputated side were compared with those obtained on the non-amputated side. Shoulder abduction was significantly higher with the Greifer (60.9˚ ± 20.3˚, $p = 0.03$) than with the Axon-Hook (39.8˚ ± 16.9˚) and also than with the NA side (37.6 ± 19.4˚, $p = 0.02$). Shoulder abduction on the NA side (37.6 ± 19.4˚) was close to that of the Axon-Hook (39.8˚ ± 16.9˚). The percentage of time spent with shoulder abduction > 60˚ during the BBT was higher with the Greifer than with the Axon-Hook or with the NA side (53.3 ± 34.4%, 17.6 ± 27.0% and 18.4 ± 34.9%, respectively), but the differences were not significant ($p = 0.15$). A significant strong negative correlation was found between shoulder abduction and wrist position with the Axon-Hook (r = -0.86; $p < 0.01$), but not with the Greifer. Manual dexterity and satisfaction did not differ significantly between the two devices. These results revealed compensatory movements, such as shoulder abduction in transradial amputees equipped with hooks, themselves influenced by the prosthetic device settings.

**Data Availability Statement:** All relevant data are contained within the manuscript. Raw data cannot be shared publicly because of French regulations.

Anonymized raw data are available from the study promotor (dpo@ottobock.fr) with an authorization from the French Data Protection Agency (CNIL). The French Data Protection Agency (CNIL) can be contacted by following this link (https://www.cnil.fr/fr/contacter-la-cnil-standard-et-permanences-telephoniques).

**Funding:** Otto Bock France (https://www.ottobock.fr/) funded the study. Euraxi Pharma (https://www.euraxi.fr/), a contract research organization, was appointed by the sponsor to collect and analyze the data. The study was conducted in full compliance with ISO 14155 (GCP). The corresponding author had full access to all the study data and had final responsibility for the study design, the decision to submit for publication and the preparation of the manuscript.

**Competing interests:** The authors have declared that no competing interests exist.

## Introduction

Upper limb amputees equipped with a prosthetic device display unnatural or unusual movements with their other joints and segments [1–4]. Strategies for carrying out tasks of daily living involve not only adapted movements of their residual limbs, but also overuse of their non-amputated limbs, and overall compensation with their whole body, creating risks of injuries [1]. As a result, musculoskeletal disorders (MSD) are more prevalent in upper limb amputees than in the general population [5]. Compensatory movements of the upper limbs, neck and trunk expose these subjects to musculoskeletal pain in the shoulders, residual limb and spine [3, 4, 6]. The two most frequent MSD observed in case of unilateral upper limb amputation are contralateral carpal tunnel syndrome and homolateral shoulder pain, which affect about 40% of amputees [7].

Several publications [2, 4, 8–10] discuss the measurement of compensatory movements while moving objects with upper limb prostheses. Three categories of compensatory strategies have been described following comparison of transradial myoelectric prosthesis users with able-bodied subjects during bimanual tasks: prepositioning of devices and objects in the workspace, posture compensations and a range of motion compensations [10].

Kinematic assessments using motion capture technologies have proven to be valuable for identifying movement strategies in the able-bodied population [9, 11, 12] and also for assessing compensatory movements in upper limb amputees [3, 8, 13–17].

Thus, it has been shown that transradial amputees with body-powered prostheses displayed relatively normal shoulder abduction/adduction when performing two standardized tasks (Cup Transfer and Pasta Box) in the standing position, but they compensated with trunk movement and decreased shoulder flexion/extension motion [17]. Among the standard tests, the BBT is widely used and is a validated timed measure of upper limb functional performance [18]. With a specific repetitive task such as the BBT, the motion of the upper limbs and trunk can be repeatedly observed and recorded. The cyclical nature of the task makes it ideal for motion capture [19].

The impact of loss of degrees of freedom (DOF) at wrist and finger levels was evaluated in able-bodied subjects, performing the BBT in the standing position by simulating the limitations induced by conventional prostheses with bracing and strapping. The simulation showed a decrease in BBT performance and an increase in shoulder abduction by about 20° throughout the test [11, 16]. This compensatory strategy may generate shoulder pain since the loss of distal DOF has been shown to increase shoulder muscles force generation [20].

It is important to limit shoulder abduction amplitude and duration because they are risk factors for shoulder pain and MSD, as shown in studies on occupational risks and posture at work in able-bodied subjects. Many studies or reports in the field of ergonomics thus give thresholds for arm elevation and/or shoulder abduction as well as thresholds for the length of time these positions can be maintained before a risk of shoulder MSD appears [21]. A Swedish research group has suggested that unsupported arm elevation > 60° for more than 10% of the workday or elevation > 30° for more than 50% of the workday could increase the risk of disorders [22].

Compensation studies have been performed with both myoelectric and body-powered prostheses [13, 15, 23]. The functionalities of the prosthetic components, such as the type of terminal effector or wrist mobility, also seem to influence upper limb movements in amputees. Wrist flexion seems to reduce shoulder and elbow compensatory movements. Transradial amputees were more satisfied and had higher functional scores performing Southampton Hand Assessment Procedure tasks with a mobile wrist in flexion compared to with a static wrist. However, the benefit provided by wrist flexion on shoulder compensation was not

always confirmed. These different results could be explained by the different prosthetic hand types that were associated with the mobile wrist.

Depending on the subject's expectations and life project, various prosthetic solutions are available with different control systems and terminal effectors. Among them, myoelectric systems provide functional and esthetic solutions when associated with a prosthetic hand. Nevertheless, some users appreciate having a myoelectric hook for certain professional or leisure tasks, because this non-morphologic tool offers more strength and precision than a prosthetic hand. Otto Bock HealthCare Products GmbH has developed two myoelectric hooks, the Greifer and the Axon-Hook. Concerning DOF, the Greifer allows radial/ulnar deviation, while the Axon-Hook allows wrist flexion/extension. Concerning shape, the Axon-Hook is less bulky than the Greifer.

No study has been published on the Axon-Hook yet, but one case report comparing the efficiency of the Greifer with three other terminal effectors when performing the BBT and bimanual tasks of daily [23].

By analogy with observations of prosthetic hands [24] and following observations of upper limb amputees in their daily lives, it seems that with the Greifer, the lack of wrist flexion, associated with limited visibility due to its volume and design, could lead to compensatory movements such as increased shoulder abduction.

In this context, this study aimed to compare compensatory shoulder movements and the percentage of time spent with shoulder abduction $> 60°$ using motion analysis in transradial amputees successively equipped with the Greifer and the Axon-Hook while performing a standardized task (BBT).

Manual dexterity (quantity of blocks moved), user satisfaction with the Greifer and Axon-Hook and hook preferences were also reported.

Shoulder abduction (amplitudes and durations) and manual dexterity with each hook on the prosthetic (P) side were also compared with the non-amputated (NA) side.

## Materials and methods

### Study design

A monocentric, comparative, open, randomized crossover trial (Fig 1) was conducted between September 2016 and February 2017 at the Regional Rehabilitation Institute of Nancy, France. The study compared the Greifer and the Axon-Hook, with each participant being his own comparator during successive sequences. The study protocol was registered with the French National Agency for Medicines and Health Products (ANSM) and approved by an independent ethics committee (Comité de Protection des Personnes Est-III) on August 29, 2016 (N˚16.07.02). Data collection complied with the reference methodology for interventional trials (MR-001) issued by the French Data Protection Agency (CNIL). The ClinicalTrials.gov Identifier of this study was NCT04522349 and registered as 'Compensatory Movements with Axon-Hook and Greifer in Transradial Amputees'. The registration on clinical trial was made after the end of the study, although not compulsory, it nevertheless turns out to be essential for publication. The authors confirm that all ongoing and related trials for this intervention are registered.

The sample size calculation was based on the expected difference between the two devices on the mean shoulder abduction during the Box and Blocks Test. Based on observations made during recordings in routine clinical practice, the difference between the two mean shoulder abduction was estimated at 25˚. To be able to detect such difference with a power of 90 using a threshold for statistical significance of 0.05, we needed to include 8 patients.

The subjects were recruited during prosthetic medical appointments usually carried out at the rehabilitation center. During the inclusion period, all persons who met the inclusion

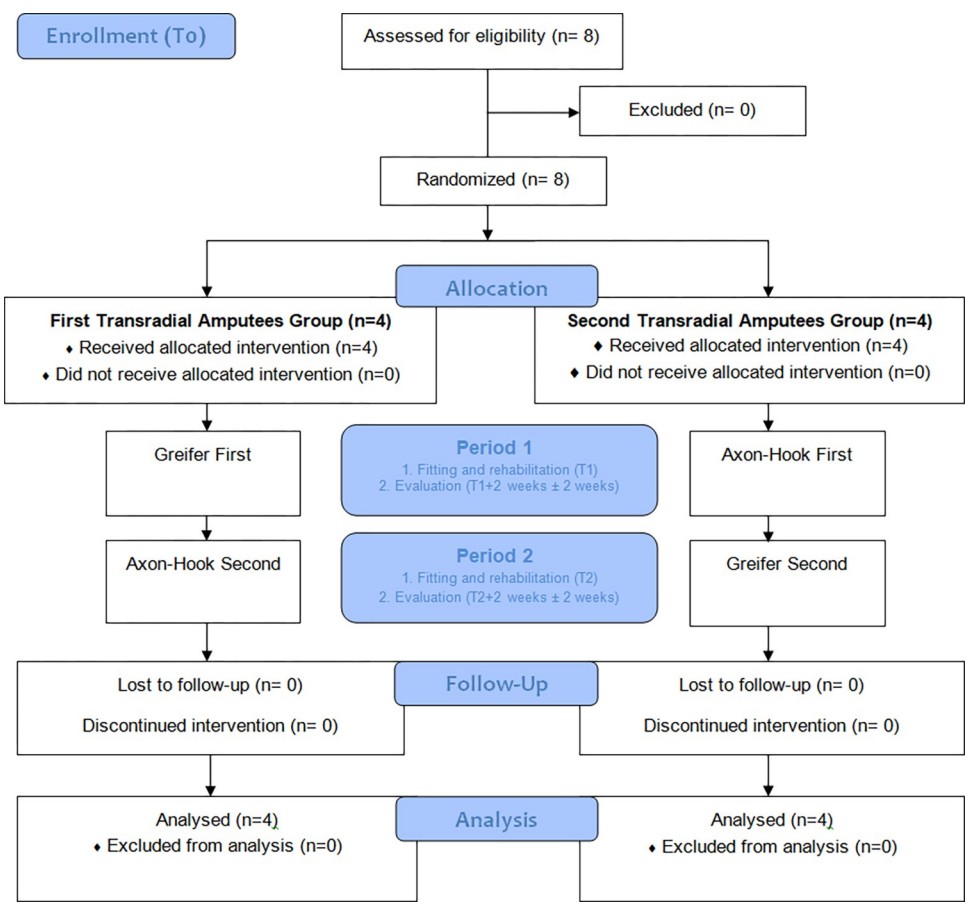

**Fig 1. Study design and patients flow diagram.**

criteria were invited to participate in the study by the investigating doctor. They received all the required oral and written information. All the subjects included in the trial signed the informed consent forms.

The participants were randomly assigned (by choosing an envelope) to one of the two study arms, which differed by the order in which the two devices were fitted (Greifer then Axon-Hook or Axon-Hook then Greifer, Fig 1). Given that the technologies actuating the two hooks are different, a copy was made of the socket, and another prosthesis manufactured for evaluation of the other hook. At T1, each participant underwent at least one rehabilitation session (according to the individual participant's needs) to ensure that the first device could be well controlled and that the patient-device unit was efficient. After that, the participants had a home trial period of two to four weeks during which they were asked to use the hook when considered relevant. A first assessment was performed at the end of this period. The participant was then fitted with the second hook (T2) and underwent similar sequences. The second rehabilitation session served as a "wash-out" period for the first hook. Each evaluation meeting lasted around one hour.

## Participants

Eligible participants were adult subjects having undergone unilateral transradial amputation due to acquired (a minimum of six month prior) or congenital causes, who regularly used a

myoelectric prosthesis and had good control over the prosthesis, whose stump was stabilized, whose professional activity or life project justified the use of a myoelectric hook, who was capable of understanding the test instructions and who have given their free and informed written consent. The exclusion criteria applied to the recruitment of participants were as follows: minors, pregnant women, persons in emergency situations, persons unable to personally give their consent, persons who are psychically or linguistically unable to understand the instructions for taking the research tests, persons not available to comply with the entire study protocol.

### Assessed devices

The Greifer is a Myobock system hook (Otto Bock HealthCare Products GmbH) that can be used as a complement to a myoelectric hand. Its major characteristics include two strong mobile fingertips, a rather voluminous design and the possibility of being manually orientated medially or laterally at the wrist level (radial or ulnar deviation). The Axon-Hook is compatible with Axon-Bus technology and can be used as a complement to the Michelangelo hand (Otto Bock HealthCare Products GmbH). It differs from the Greifer in that it allows flexion/extension wrist, it has thin fingertips with one being fixed (Fig 2).

### Data collection and analysis

**Box and Blocks Test.**   The Box and Blocks Test (BBT) is a manual dexterity test that consists in moving, one by one, as many cubic wooden blocks (2.5 cm in size) as possible in one minute from one compartment of a box to the other (dimensions 53.7 cm X 25.4 cm X 8.5 cm). The BBT score corresponds to the number of blocks moved [18]. A high score shows good manual dexterity. Prior to each test and after training with a few blocks, the participant could choose the most suitable wrist setting for the hook according to him–wrist radial/ulnar deviation with the Greifer and flexion/extension with the Axon-Hook. Before each evaluation, this freely chosen position was measured and recorded. Each participant could also choose whether or not to use the motorized wrist rotation: the choice was retained for both evaluations.

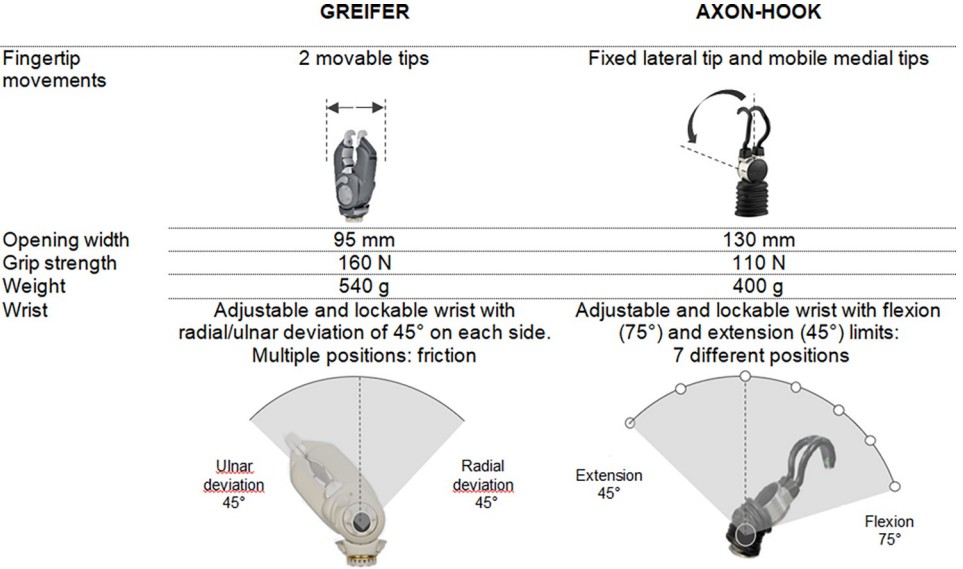

**Fig 2. Characteristics of the Greifer and the Axon-Hook.**

The non-amputated side was also assessed during the first evaluation. This repetitive test is coherent with tasks that are performed with a hook and is compatible with the records of motion analysis laboratories.

**Kinematic data.**   An eight-camera Vicon^TM motion capture system (Vicon Motion Systems Ltd., Oxford, UK) was used to collect kinematic data during the BBT. Prior to data collection, the system was calibrated according to the manufacturer's guidelines. Data were collected from the optical cameras at a rate of 100 Hz. Reflective markers were placed on each participant's head, trunk, upper limbs and pelvis according to the Plug-In-Gait upper body model from Vicon^TM (Fig 3). On the non-amputated hand, the marker was placed just below the third metacarpal bone and on the prosthetic side, it was placed similarly on both hooks. All markers were attached using hypo-allergenic double-sided tape. To visualize the position of the objects, the plane of work and define sequences, four markers were placed on each corner of the table and three were placed on the BBT set.

Two digital video cameras were synchronized with an optoelectronic system to record each participant's BBT session from the front and side.

Each participant was instructed to sit on a stool in front of a height-adjustable table on which the BBT set was placed (Fig 4). Masking tape was stuck on the table so the set would always be placed in the same position. The table was adjusted so that the participant's elbow flexion at rest was 90° with the forearm flat on the table. A short (2 seconds) static subject calibration recording was made prior to data collection to apply Plug-In-Gait upper body model. Individual anthropometric parameters (height, weight, distance between the anterior superior iliac spines, hand thickness, wrist/elbow width, shoulder offset) were used to estimate the position of each joint center. Recordings were then made of the dynamic sessions.

Shoulder kinematics were collected in the three-dimensional plane for the Prosthetic (P) side and for the Non-Amputated (NA) side. Initialization of movement was defined by the first displacement of the hook or hand marker on the side performing the test. Mean motion and standard deviations were calculated. The percentage of BBT time with shoulder abduction values > 60° was also determined.

**Satisfaction.**   Participant satisfaction was assessed using the French version of the validated and frequently used Evaluation of Satisfaction with Assistive Technology (ESAT 2.0) [25, 26]. It comprises 8 items related to the technology and 4 items related to services, rated from 1 (least satisfied) to 5 (best satisfied). Users determine the three most important items for themselves. The global score is the mean of the 12 items. Two subscores can be calculated, for technology and for services.

**Preferences.**   At the end of the second evaluation, each participant was asked which hook he preferred.

Adverse events were reported throughout the study.

Data were collected using paper report forms and questionnaires, then entered in an electronic database. Data management activities were carried out by a clinical research organization compliant with good clinical practices.

## Statistical analysis

Quantitative data were reported as the mean (Standard Deviation). We performed mixed-design analyses of variance (ANOVA) with "Group" (Axon-Hook then Greifer, or Greifer then Axon-Hook) as the between-participant factor and the "Hand" (Axon-Hook, Greifer or Non-Amputated side) as the within-participant factor. When one factor was statistically significant, and when the interaction between "Group" and "Hand" was statistically significant for a given variable, size effects was reported as the partial eta 2 ($\eta_p 2$), and the Fisher post-hoc. When the interaction effect was not significant, it was dropped out from the analysis and only

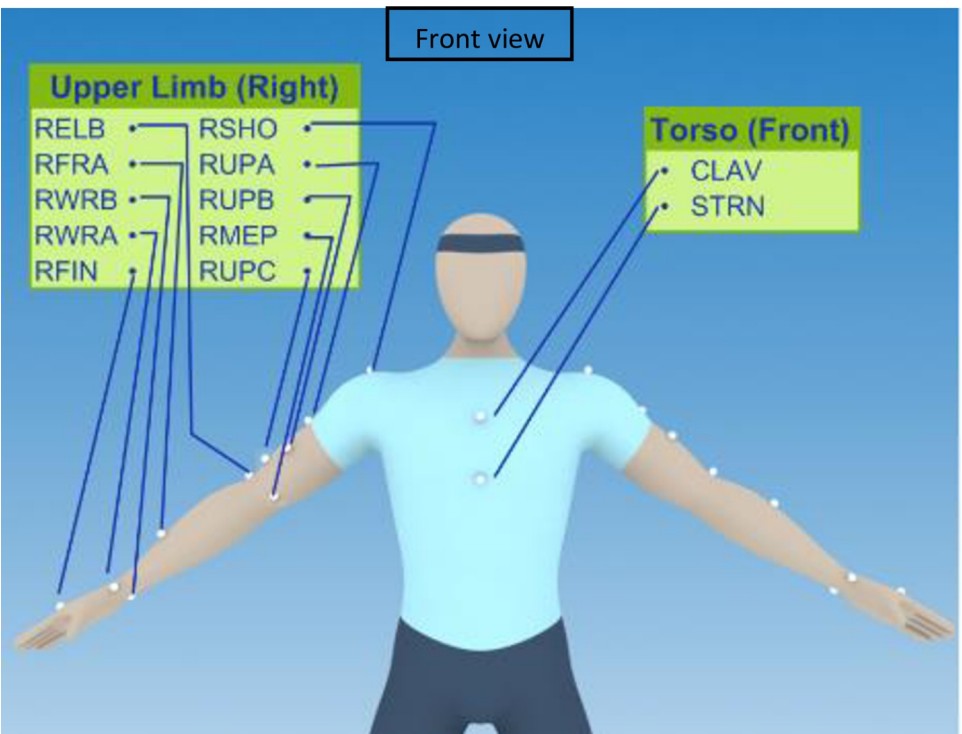

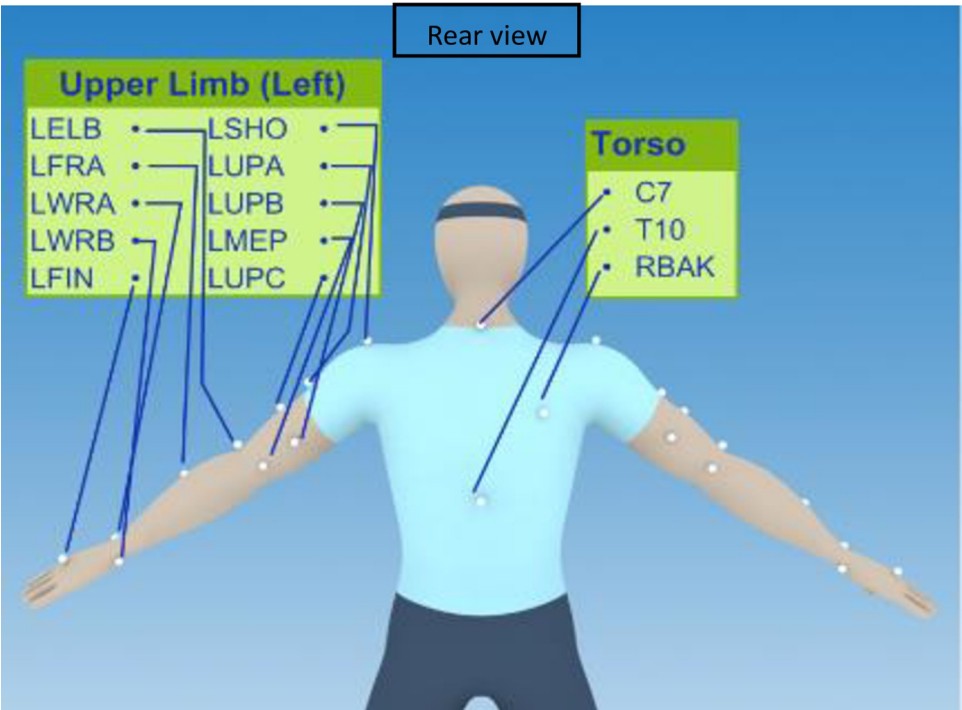

**Fig 3. Position of the Plug-In-Gait upper body model markers.**

the main effects were tested. The threshold for statistical significance was set to $\alpha = 0.05$. All statistical analyses were performed using Statistica software (version 13, Tibco Software Inc., Palo Alto, CA, USA).

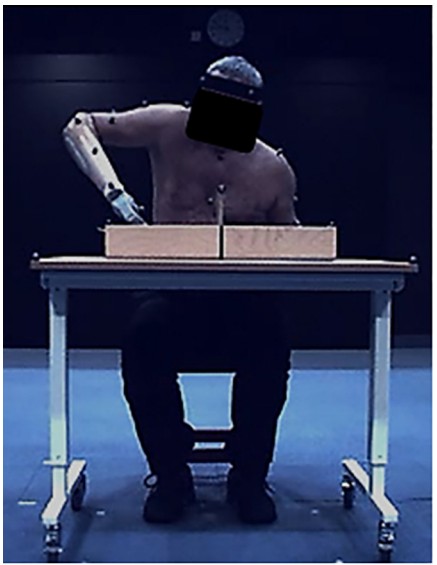
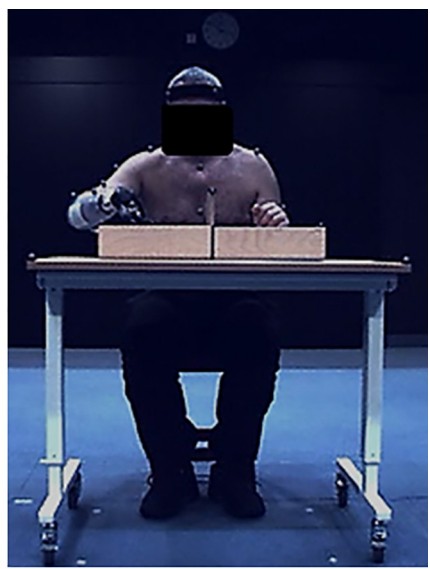

BBT with the Greifer                        BBT with the Axon-Hook

**Fig 4. BBT with the Greifer and the Axon-hook.** Pictures showing a transradial amputee assessed while performing the BBT in a motion analysis laboratory during the pilot phase.

A post-hoc analysis was performed to assess the association between participants' chosen wrist adjustment and mean shoulder abduction during BBT by calculating the Spearman correlation coefficient and $p$-values.

The analysis was planned on the intention to treat population. P-value less than 0.05 were considered statistically significant.

## Results

### Characteristics of the participants

Eight subjects (all male, mean age: 44.8 ± 15.8 years) were enrolled and randomized to the 2 groups (n = 4 for each group). All the subjects met the inclusion criteria, strictly followed the protocol and were assessed with both devices. Therefore, all participants were treated per protocol and belonged to the intention to treat set. No adverse events were reported by any of the participants.

The characteristics of the participants are summarized in Table 1. Seven participants reported having leisure activities, including activities that could justify the use of a hook. None had an associated disability. Their non-amputated limbs were healthy. Half the participants were already using a Greifer before the study. In five subjects, wrist rotation was motorized. The participants' characteristics were similar in both randomization groups.

### Shoulder abduction

There was no interaction effect between the "Hand" and the "Group" (F = 0.25, $p$ = 0.78) on the shoulder abduction. After removing the interaction effect from the analysis, it appears that there was no "Group" effect (F = 0.09; $p$ = 0.95) and a main effect of the "Hand" (F = 3.96; $p$ = 0.04; $\eta_p^2$ = 0.36). Post-hoc analysis revealed that the shoulder abduction was significantly lower with the Axon-Hook (39.8 ± 16.9˚) than with the Greifer (60.9 ± 20.3˚) ($p$ = 0.03) (Fig 5). Moreover, there was no significant difference between the Axon-Hook and the NA side

**Table 1. Characteristics of participants.**

| # | Sex | Age (years) | Laterality | A side | Level of TRA | Time since A (years) | Etiology | Type of work | Leisure activities | Greifer user | Wrist rotation | Greifer wrist setting (°) | Axon-Hook wrist setting (°) | Group |
|---|---|---|---|---|---|---|---|---|---|---|---|---|---|---|
| 1 | M | 63 | RH | Left | S | 12.2 | T | Retired | Walking Manual activities Woodwork Hunting Taking care of grand-children | No | M | RD = 35 | F = 20 | G-A |
| 2 | M | 43 | RH | Right | S | 17.3 | T | Storekeeper | Cycling Hiking Manual activities Gardening Woodwork | Yes | M | UD = 10 | NP = 0 | G-A |
| 3 | M | 26 | RH | Right | S | 0.7 | T | None | Soccer | No | M | RD = 10 | F = 30 | A-G |
| 4 | M | 45 | LH | Right | M | 45.2 | C | Electrician | Running Mountain biking Manual activities Gardening | No | NM | RD = 20 | F = 15 | A-G |
| 5 | M | 52 | RH | Right | M | 3.4 | T | Farmer | - | Yes | NM | RD = 30 | NP = 0 | A-G |
| 6 | M | 23 | RH | Right | L | 1.7 | T | None | Cycling Paragliding Manual activities | No | NM | RD = 40 | F = 15 | G-A |
| 7 | M | 67 | RH | Left | L | 49.4 | T | Retired | Swimming Cycling Karate Marquetry Jewelry making | Yes | M | RD = 20 | F = 20 | G-A |
| 8 | M | 39 | RH | Left | S | 10.4 | T | Caretaker | Model building Blacksmithing Leather work Hiking Rifle shooting | Yes | M | RD = 40 | F = 40 | A-G |

Laterality: RH = right-handed: LH = left-handed: A side = amputated side

Level of TRA = transradial amputation: S = short (upper 1/3): M = medium (middle 1/3)

L = long (lower 1/3)

Time since A = time since amputation

Etiology: T = traumatic: C = congenital

Wrist rotation: M = motorized: NM = non-motorized

Greifer wrist setting: RD = radial deviation: UD = ulnar deviation

Axon-Hook wrist setting: F = flexion: NP = neutral position

Group (after randomization): G-A = Greifer then Axon-Hook: A-G = Axon-Hook then Greifer

$(37.6 \pm 19.4°)$ ($p = 0.82$). Lastly, the shoulder abduction with the Greifer was significantly higher than with the NA side ($p = 0.02$).

There was no interaction effect between the "Hand" and the "Group" (F = 0.24; $p = 0.78$) on the percentage time spent with shoulder abduction $> 60°$, no more than "Hand" (F = 2.44; $p = 0.12$) or "Group" (F = 0.15; $p = 0.92$) effect after removing the interaction effect from the statistical analysis (Fig 6). The mean values were $53.3 \pm 3.4\%$ with the Greifer, $17.6 \pm 27.0\%$ with the Axon-Hook and $18.4 \pm 34.9\%$ with the NA side.

When assessing the effect of wrist settings, a significant strong negative correlation was found between shoulder abduction and flexion with the Axon-Hook (r = -0.86; $p < 0.01$). The correlation between shoulder abduction and radial deviation with the Greifer was weak and not significant (r = 0.38; $p = 0.34$).

When examining individual results, we observed that one participant (#2) had an original compensatory strategy for grabbing and moving the blocks, with lower mean shoulder abduction with the Greifer than with the Axon-Hook (Fig 7). With the Greifer (Fig 8, left picture), he placed the prosthetic side against his trunk, elevating his shoulder stump while tilting his trunk laterally to the non-amputated side. The strategy was not based on shoulder abduction in contrast to the other participants. It is important to note that participant #2 had selected a

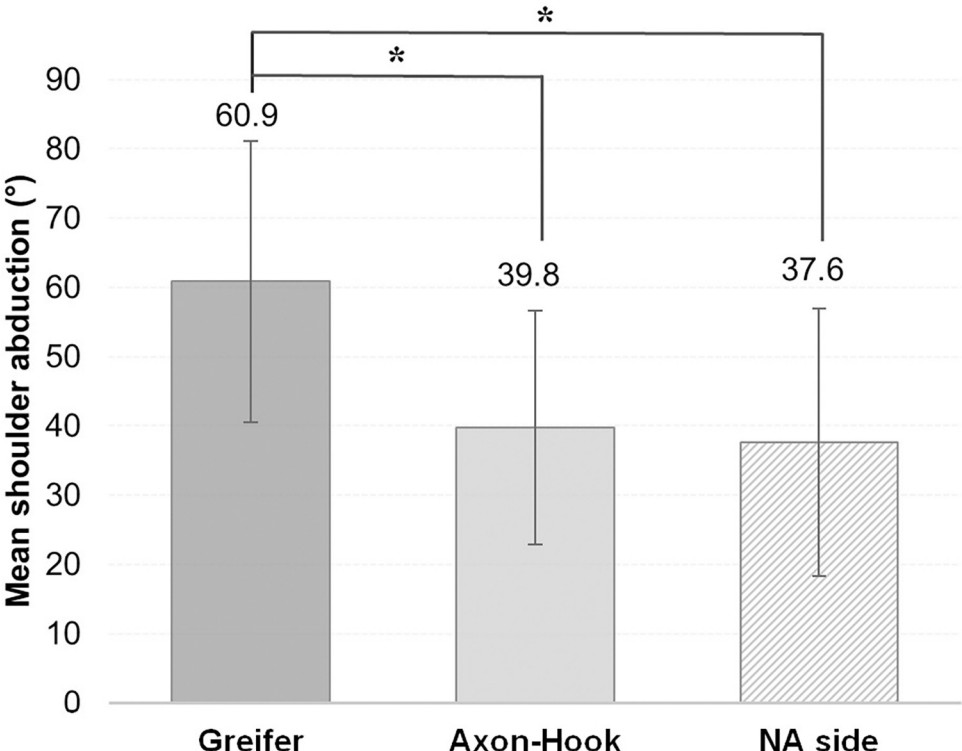

**Fig 5. Shoulder abduction with the Greifer, the Axon-Hook and the NA side during the BBT.** Significant (*$p < 0.05$) higher shoulder abduction with the Greifer than with the Axon-Hook and than with the NA side.

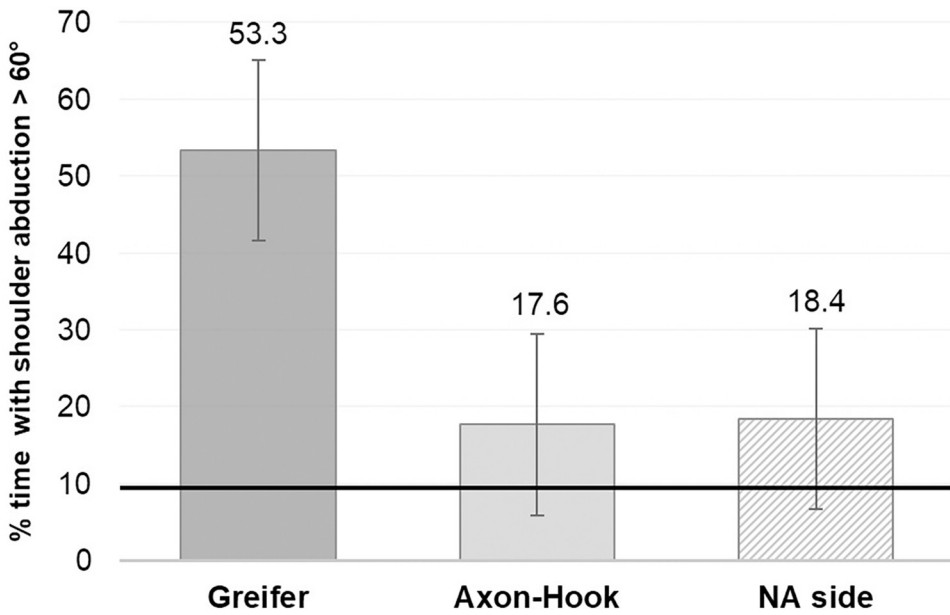

**Fig 6. Time with shoulder abduction > 60˚ with the Greifer, the Axon-Hook and the NA side during the BBT.** The thick black line indicates the threshold of 10%, considered as the percentage of time in a workday above which patients are at risk of developing MSD. There was no statistical difference between the groups ($p = 0.92$ for mixed-design analysis of variance).

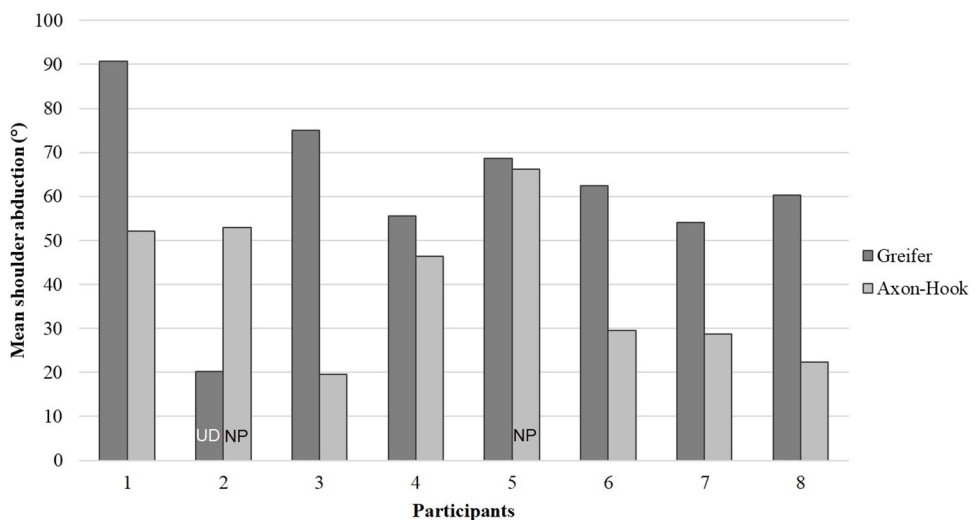

**Fig 7. Shoulder abduction for each participant, with the Greifer and the Axon-Hook during the BBT.**

10˚ ulnar deviation wrist setting with the Greifer and no flexion with the Axon-Hook, which was different from most of the other participants (Table 1).

## Manual dexterity

There was no interaction effect between the "Hand" and the "Group" (F = 0.37; p = 0.69). After removing the interaction effect from the analysis, it appears that there was no "Group" effect (F = 0.3; p = 0.82) and a main effect of the "Hand" (F = 90.32; $p < 0.001$; $\eta_p^2 = 0.92$).

Post-hoc analyses revealed a higher score with the NA side (7.4 ± 6.2 blocks) compared to the Axon-Hook (25.4 ± 10.0 blocks, $p < 0.001$) and to the Greifer (23.9 ± 8.6 blocks, $p < 0.001$) (Fig 9).

## Prosthetic user satisfaction

Responses to the ESAT 2.0 questionnaire items showed that global satisfaction with the Greifer and the Axon-Hook were not significantly different (4.39 ± 0.26 *versus* 4.43 ± 0.52; p = 0.80).

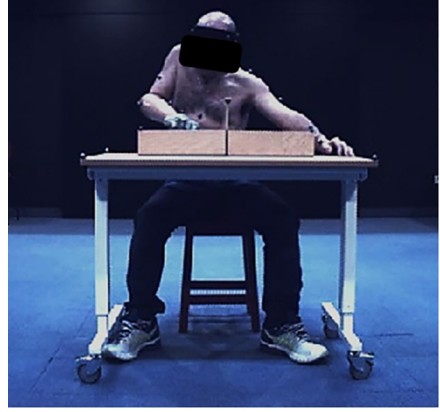

BBT with the Greifer                    BBT with the Axon-Hook

**Fig 8. Atypical compensatory strategy of participant #2.**

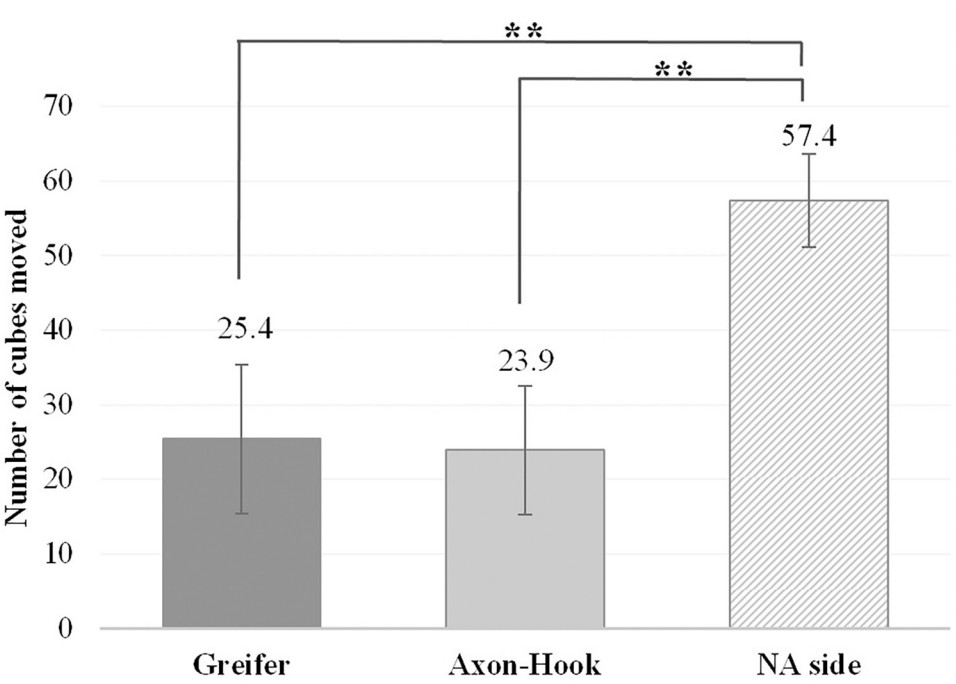

**Fig 9. BBT scores with the Greifer, the Axon-Hook and the NA side.** Main effect of the "Hand" (F = 90.32; $p < 0.001$; $\eta_p^2 = 0.92$) with a significant greater score with the NA side compared to the Axon-Hook (**$p < 0.001$) and to the Greifer (**$p < 0.001$). No significant difference was observed between the Greifer and the Axon-Hook ($p = 0.60$).

With the Greifer, the most important criteria cited were safety (n = 5), durability (n = 5) and effectiveness (n = 5). With the Axon-Hook, participants stated that the most important criteria were weight (n = 6) and efficiency (n = 5), with solidity and ease of use being ranked equally (n = 4).

## Prosthetic user preference

At the end of the study, six participants (75%) stated they preferred the Axon-Hook while two (25%) said they preferred the Greifer.

## Discussion

The main purpose of this study was to compare kinematic compensatory shoulder movement and dexterity in transradial amputees successively equipped with a Greifer and an Axon-hook during a BBT session.

Mean shoulder abduction obtained during the BBT sessions was significantly greater with the Greifer compared to the Axon-Hook and with the Greifer compared to the NA side. The difference of percentage of time with shoulder abduction > 60˚ was more than 30% higher between the Greifer and the two other sides (Axon-Hook and NA side), but were not statistically significant. For these shoulder abduction variables, the difference between the Axon-Hook and the NA side was less than 1 degree (amplitude) and less than 1% (% of time with shoulder abduction > 60˚) suggesting that the second hook could be more appropriate for limiting compensatory shoulder abduction. Although the differences were not statistically significant for percentage of time with shoulder abduction > 60˚ (probably due to the small number

of participants and the presence of the atypical subject) these values were nonetheless of clinical relevance. This finding needs to be confirmed with a larger cohort performing the BBT.

Although it is recommended that the BBT be performed in the sitting position, as was the case in this study, the literature only contains shoulder data with the BBT being performed in the standing position. In 19 standing, able-bodied subjects, shoulder abduction ranged between 15 and 20˚ when they were not restricted, and up to 35˚ when wrist and [11, 16]. Non-restricted values were far lower than those observed on the NA side (37.6˚), whereas restricted data were in line with the data observed with the Axon-Hook (39.7˚). The sitting position is probably more constraining than the standing position, and compensations via movable body parts (at the trunk-head-upper limb level), including shoulder abduction, are even more necessary. In the standing position, trunk inclination was favored [11, 16]. Relatively high shoulder abduction on the non-amputated side in our study could reflect a search for symmetry between the two sides by the amputees as has already been observed in lower limb amputees during walking [27] and in upper limb amputees performing reachability judgement tasks [28]. It can be assumed that concern for symmetry is exceeded by the need for compensatory movement when participants are fitted with the Greifer.

An absence of sensory feedback and limited DOF with prostheses could be the first explanation of the compensatory movements that open the way for MSD. Secondly, the type of DOF (i.e., the rotation axis around which the movements are possible) allowed by the two hooks could explain the shoulder abduction differences. The Greifer allows wrist radial/ulnar deviation, whereas the Axon-Hook allows wrist flexion/extension, which is more suitable for a prehensive movement such as that used for the BBT. It is already known that with prosthetic hands, wrist flexion improves gesture efficiency and limits compensations [29]. We showed a significant strong negative correlation between wrist flexion and shoulder abduction with the Axon-Hook: greater wrist flexion is associated with less shoulder abduction. These data suggest that terminal effectors designed with higher degrees of freedom in flexion are valuable in that they limit compensatory abduction movements of the shoulder during prehensive tasks. Wrist flexion could therefore also limit shoulder abduction with non-morphometric end effectors even if the type of grip is different from morphometric effectors (inter-finger grip *versus* tri-digital grip).

Conversely, with the Greifer, the ulnar or radial inclination of the wrist does not seem to influence shoulder abduction. The type of grip may also affect compensation: the Greifer has two movable fingertips while the Axon Hook has one movable and one stationary fingertip that may serve as a fixed support point to grab the block.

The significant greater shoulder abduction observed with the Greifer may be explained by the strategy used by the participants to compensate with their shoulder when using the prosthetic limb.

The question may be raised whether the wrist setting of the Greifer (ulnar or radial deviation) affects trunk compensation. The only participant (#2) having selected ulnar inclination showed predominant trunk compensation.

Most participants (except #2) chose radial deviation, which indeed seems more appropriate since this hook orientation makes it easier to approach the blocks and corresponds to the movement imposed by the BBT (moving of blocks from the P side to the NA side).

Irrespective of the side, the percentage of time spent with shoulder abduction > 60˚ during the BBT was above the threshold of 10% beyond which there is a risk of MSD [22]. With the Greifer, this position was maintained for more than half of the working session, whereas the recommendation for avoiding MSD is to maintain abduction below 30˚.

Significant greater shoulder abduction could be a source of major shoulder strain that could, in the long-term, cause pain in this joint [30]. Prolonged arm elevation may result in

shoulder pain and MSD at various thresholds: 10% of the workday with shoulder abduction $> 60°$ triggers MSD [22]; more than 2h/day with shoulder abduction $> 60°$ is associated with shoulder pain [31]. By its repetitive aspect, the BBT accentuates the repercussions of compensatory shoulder movement beyond the usual functional limits of the joint. In real life conditions, prosthetic users are confronted with various contexts and grip situations that cannot be appraised in a BBT session. Modified versions of the BBT, in which the placement and order of the blocks are imposed, were developed to be closer to real life situations [11, 13, 32]. The choice of the standard BBT in this protocol was intentional, to allow each participant adopt his own movement strategy, so as to observe the different compensatory strategies engendered by the prosthetic device and guided by the participant's choices and needs.

The study was carried out over a short period, but the repercussions of compensatory movements (MSD, pain, injuries, etc...) only become visible after long periods of cyclic, repetitive movements. It would be interesting to follow the evolution of strategies over time on both sides (P side and NA side) and to check for the onset of MSD. Shoulder compensation is a risk on both sides. On the prosthetic side, it seems that compensatory movements are unavoidable regardless of the device used (based on currently available devices). The major concern of rehabilitation care is to help patients use the most effective and least harmful unavoidable compensatory strategy and to abandon avoidable ones. In order to do this, we need to improve our understanding of compensatory movement mechanisms and their causes. During rehabilitation, it seems essential to detect and correct avoidable shoulder compensatory strategies on the NA side that increase the risk of MSD.

This study included an atypical participant (#2) who had a specific compensatory strategy. This case well illustrates the multi-articular/segmental complexity of compensatory strategies and emphasizes the importance of evaluating global compensatory strategies when carrying out a motor task. As pointed out by Metzger [4], it seems that in the performance of the activities of daily living (dressing), some transradial amputees tilt their trunk laterally, which is an energy-intensive strategy, regardless of the technical limits of the prosthetic devices or residual joints. During rehabilitation, and particularly in the definition phase of the device project, it would be interesting to help patients think differently about the possibilities and ways of using their prostheses, and to integrate the patients' feedback on the use of the devices, in order to increase the efficiency of the patient-prosthesis unit.

Functional capacity or manual dexterity, reflected by the number of blocks moved during the BBT, was not significantly different with the Greifer and the Axon-Hook, either from a statistical or a clinical point of view, as the difference of $1.50 \pm 1.54$ blocks was below the Minimum Detectable Change (MDC90 is set at 6.49 blocks for upper limb amputees [33]). Conversely, the BBT scores with the Greifer or the Axon-Hook were more than two times lower than with the NA side, with a far higher difference than the MDC90, indicating that a prosthesis cannot fully compensate for a patient's disability.

In the literature, the standard BBT had already been used to assess manual dexterity in upper limb amputees. Only one study evaluated the Greifer during a BBT session in three transradial amputees and reported a mean score of 25.6 blocks, in line with our results (25.4 blocks) [23]. Lower BBT scores, however, were reported in two studies assessing unilateral amputees using myoelectric hands instead of hooks: 20.9 blocks (N = 17) [34] and 14 blocks (N = 8) [35]. In these two studies the participants were also evaluated performing the BBT in a sitting position, but they were fitted with myoelectric hands, and not with hooks like in our study. The higher scores obtained in our study could therefore be explained by the type of prosthetic terminal effector. Indeed, hook fingertips probably allow a faster approach and more effective picking-up of the block than myoelectric hands. Myoelectric hands are therefore probably less functional, which also appears to be the case with body-powered hands.

Haverkate et al. assessed manual dexterity in able-bodied subjects wearing a forearm prosthesis simulator with body-powered end effectors, i.e., the Ottobock® Hand, a flat hook, and a hook with two fingertips pointing down. The BBT scores were significantly different: 17, 22 and 29 blocks, respectively [36]. These data show that manual dexterity is better with body-powered hooks than with body-powered hands. Hooks seem more convenient for performing repetitive grip tasks because they allow better manual dexterity. However, as the study tested prosthetic simulators on able-bodied volunteers' upper limbs, the results were only an approximation of real performance.

Kontson et al., in turn, used a wrist brace and finger straps in able-bodied subjects (limiting wrist radial/ulnar deviation, wrist flexion/extension, finger flexion, and restricting finger use to index and middle-finger) to simulate prosthesis DOF restrictions during BBT sessions in the standing position. In these conditions, the mean BBT score was 45 blocks, *versus* 60 blocks for the controls [16]. Degrees of freedom therefore appear to influence manual dexterity. In our study, the fact that there was no significant difference in manual dexterity between the two hooks can be explained by equivalent, although different DOF. Test performances are clearly better in a restrictive situation in able-bodied subjects than in transradial amputees fitted with prostheses as in our study. This can be explained by the constraints of prosthetic control and loss of sensory feedback in amputees, which remains present in able-bodied subjects.

Beyond the fact that in this study there were statistical differences in compensatory movements (shoulder abduction significantly higher with Greifer than with Axon Hook), 75% of the study participants said they preferred the Axon-Hook. The notion of preference was based on multiple factors that contributed to the efficiency of the patient-prosthesis unit in the patient's overall life project. A more detailed analysis of the reasons justifying hook preference should be carried out and factored into the prosthetic project.

The rehabilitation program focusing on compensatory movements of prosthetic users should be personalized and should take into account this notion. A better understanding of compensatory strategies that would help discriminate between useful/unavoidable and harmful/avoidable ones, would allow more efficient rehabilitation care in upper limb amputees and improve their autonomy with respect to prosthetic settings. It is important that compensatory movements on the non-amputated side also be prevented.

This search for more efficient compensatory strategies followed by their automation is delicate since compensatory strategies are multifactorial and depend on the patient (medical history, experience, esthetic choices. . .), on the prosthesis (components, settings. . .), on the patient-prosthesis interaction, on rehabilitation, but also on the type and complexity of the motor task needing to be performed.

## Additional considerations and study limitations

Some participants had already used the Greifer in their daily lives, which may have influenced results regarding manual dexterity. Nevertheless, the potential impact of motor learning on the limitation of shoulder abduction has not been demonstrated at all, quite the contrary.

All participants had different prosthetic settings. It would be interesting to impose the same setting on all participants and to evaluate different setting combinations to better understand the influence of settings on compensatory movements.

The kinematic study focused on the shoulder, although other joints are involved in the overall compensatory strategy.

The data interpretation following short BBT sessions must be nuanced: the BBT, although convenient and easy to implement, is not extrapolable to daily living situations and a full working day.

Our study focused on shoulder kinematics. Future research could complete this work by studying trunk, elbow and head kinematics in a multiplane approach, to better describe global compensatory strategies.

## Conclusion

This study showed that transradial amputees fitted with hooks mainly use shoulder abduction as a compensatory movement during the BBT functional capacity test. Mean abduction amplitudes were significantly lower with the Axon-Hook than with the Greifer and time spent above 60˚ was also lower with the Axon-Hook than with the Greifer, but not significantly for this variable. The higher amplitudes and durations of shoulder abduction with the Greifer are important variables that must be taken into consideration because they provide information on the risk of developing MSD in transradial amputees.

This study showed that the effect of settings on compensatory shoulder movements not only concern prosthetic hands, but also non morphometric end effectors. Manual dexterity was similar with both hooks, but relatively poorer than with the non-amputated hands. Global satisfaction scores were also similar with both hooks, even though 6 of the 8 participants declared they preferred the Axon-Hook. Further research on compensatory strategies and end effector specifications would help adapt rehabilitation programs, optimize patient-prosthesis interactions and improve the autonomy and quality of life of amputees.

## Supporting information

**S1 Checklist. Consort checklist.**
(DOC)

**S1 Protocol. Official protocol (English version).**
(DOCX)

**S2 Protocol. Official protocol (French version).**
(DOCX)

## Acknowledgments

This paper and the clinical trial on which it is based it would not have been possible without the commitment and professionalism of all the health professionals and stakeholders from the Regional Institute of Rehabilitation in Nancy, who were involved in the study. Special thanks are extended to Marie-Agnès Haldric and Nathalie Rodhain for the registration, welcoming and support of the participants.

The authors would also like to thank Odile Capronnier for her contribution to the review and formatting of the manuscript, and Jeanne Beattie for the linguistic review.

## Author Contributions

**Conceptualization:** Amélie Touillet, Noël Martinet, Isabelle Loiret, Jean Paysant.

**Data curation:** Amélie Touillet, Constance Billon-Grumillier, Jonathan Pierret.

**Formal analysis:** Amélie Touillet, Constance Billon-Grumillier, Jonathan Pierret.

**Funding acquisition:** Noël Martinet, Jean Paysant.

**Investigation:** Amélie Touillet, Constance Billon-Grumillier, Isabelle Loiret.

**Methodology:** Amélie Touillet, Constance Billon-Grumillier, Isabelle Loiret.

**Project administration:** Amélie Touillet, Noël Martinet, Isabelle Loiret, Jean Paysant.

**Resources:** Amélie Touillet, Constance Billon-Grumillier.

**Software:** Constance Billon-Grumillier, Jonathan Pierret.

**Supervision:** Amélie Touillet, Noël Martinet, Isabelle Loiret, Jean Paysant.

**Validation:** Amélie Touillet, Constance Billon-Grumillier.

**Visualization:** Amélie Touillet, Constance Billon-Grumillier, Jonathan Pierret.

**Writing – original draft:** Amélie Touillet, Constance Billon-Grumillier, Jonathan Pierret, Pierrick Herbe.

**Writing – review & editing:** Amélie Touillet, Constance Billon-Grumillier, Jonathan Pierret, Pierrick Herbe.

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
