## [Decision Letter · Decision Letter 0]

20 Oct 2021

PONE-D-21-05606Comparison of compensatory shoulder movements, functionality and satisfaction in transradial amputees fitted with two prosthetic myoelectric hooksPLOS ONE

Dear Dr. BILLON-GRUMILLIER,

Thank you for submitting your manuscript to PLOS ONE. After careful consideration, we feel that it has merit but does not fully meet PLOS ONE’s publication criteria as it currently stands. Therefore, we invite you to submit a revised version of the manuscript that addresses the points raised during the review process.

We look forward to receiving your revised manuscript.

Kind regards,

Bijan Najafi

Academic Editor

PLOS ONE

Journal Requirements:

2. In your Methods section, please provide additional information about the participant recruitment method and the demographic details of your participants. Please ensure you have provided sufficient details to replicate the analyses such as a description of any exclusion criteria that were applied to participant recruitment.

3. Please ensure you have clearly identified primary, secondary, and any exploratory outcomes.

4. Please ensure your patient flow diagram is included as Figure 1 in the main manuscript.

a. FZB(EO)19Febr21: Registration done retrospectively (after enrollment of participants)

b. FZB(EO)19Febr21: borderline ethics statement (IRB not named)

A. Thank you for submitting your clinical trial to PLOS ONE and for providing the name of the registry and the registration number. The information in the registry entry suggests that your trial was registered after patient recruitment began. PLOS ONE strongly encourages authors to register all trials before recruiting the first participant in a study.

i) your reasons for your delay in registering this study (after enrolment of participants started);

ii) confirmation that all related trials are registered by stating: “The authors confirm that all ongoing and related trials for this drug/intervention are registered”.

B. Thank you for including your ethics statement:  "The study protocol was registered with the French National Agency for Medicines and Health Products (ANSM) and approved by an independent ethics committee (CPP Est-III) on August 29, 2016 (N°16.07.02). Data collection complied with the reference methodology for interventional trials (MR-001) issued by the French Data Protection Agency (CNIL).

The subjects were recruited during prosthetic medical appointments usually carried out at the rehabilitation center. During the inclusion period, all persons who met the inclusion criteria were invited to participate in the study. They received all the required oral and written information. All the subjects included in the trial signed the informed consent forms.".   

6. Please upload a new copy of Figure 3 as the detail is not clear. Please follow the link for more information: " ext-link-type="uri" xlink:type="simple">https://blogs.plos.org/plos/2019/06/looking-good-tips-for-creating-your-plos-figures-graphics/"
" ext-link-type="uri" xlink:type="simple">https://blogs.plos.org/plos/2019/06/looking-good-tips-for-creating-your-plos-figures-graphics/"

7. Please include captions for your Supporting Information files at the end of your manuscript, and update any in-text citations to match accordingly. Please see our Supporting Information guidelines for more information: http://journals.plos.org/plosone/s/supporting-information

Reviewers' comments:

Reviewer's Responses to Questions

**Comments to the Author**

1. Is the manuscript technically sound, and do the data support the conclusions?

Reviewer #1: No

Reviewer #2: Yes

2. Has the statistical analysis been performed appropriately and rigorously? 

Reviewer #1: No

Reviewer #2: Yes

3. Have the authors made all data underlying the findings in their manuscript fully available?

Reviewer #1: No

Reviewer #2: Yes

4. Is the manuscript presented in an intelligible fashion and written in standard English?

Reviewer #1: Yes

Reviewer #2: Yes

5. Review Comments to the Author

Reviewer #1: A randomized crossover clinical trial was conducted which aimed to compare shoulder abduction, manual dexterity and satisfaction between two prosthetic myoelectic hooks. Evaluations on the unaffected arm served as a control. The results are unclear.

Major revision:

Testing the primary objectives with an all-inclusive linear mixed model would be superior to the analysis approach which has been taken. Mixed models offer the ability to test for differences in the effect of Greifer, Axon-Hook, and non-affected side. T-tests should not be needed. Additionally, specify the underlying covariance structure used in the mixed models and the criteria for selecting it.

Minor revisions:

1- Abstract: Replace the rho symbol with r since rho is the population correlation and r represents the estimate of rho.

2- Sample size justification:

a. Lines 137-8: Replace “average” with the standard statistical terminology of “mean”.

b. Specify that 25 represents the standard deviation.

State the statistical testing method, the alpha level, and the power.

3- Line 238-9: Specify the “customary descriptive statistics.”

4- Line 239: Possible typographical error: Qualitative variables were described by frequencies and percentages.

5- Line 247: Consider replacing the phrase, “The alpha-risk was set at 5%,” with “P-values less than 0.05 were considered statistically significant.”

6- Throughout the manuscript: Replace average with mean.

7- The p-value associated with a correlation is a test of the null hypothesis: correlation equal to zero; however, the absolute magnitude of the coefficient indicates the strength of the linear relationship between two variables. In general, the strength or correlation coefficient is the more important statistic to focus on.

Below is a table for interpreting correlation coefficients:

Coefficient (absolute value) Interpretation

0.90 - 1.0 Very Strong

0.70 - 0.89 Strong

0.40 - 0.69 Moderate

0.10 - 0.39 Weak

less than 0.10 Negligible correlation

Reviewer #2: This paper presents an interesting research study on two prosthetic myoelectric hooks. The paper provides a good introduction and background for the study conducted. Authors have clearly explained the method, procedure, data collection and analysis of the research study conducted on eight transradial amputees. The results obtained from the study have been statistically analyzed and thoroughly discussed in the paper. Also it is good that authors have mentioned study limitations and future considerations of their research work.

However, research output from this paper seems to be poor for several reasons. Those reasons and questions for authors are as follows;

1) In the paper it is not clear, why the authors have limit themselves just for 2 myoelectric prosthetic devices. Also it seems that conducting this research study limiting just for 2 devices will not provide substantial research outcome.

2) It is acceptable for having eight participants for this study. But as mentioned in the paper, majority (or some) participants had already being used one of the prosthetic devices that have been considered in this study. So it brings biased experimental output and decrease the quality of research outcome.

3) In my point of view, for comparisons like theses, just one experimental test is not enough (i.e. BBT). You can obtain results from multiple test and then thoroughly analyzed. This will bring good research outcome for the audience.

4) Lastly, it is good to consider several performance metrics, without being stuck into one performance metric (i.e. number of blocks moved within given time) or performance metric within the same aspect(i.e. performance at shoulder movements)

6. PLOS authors have the option to publish the peer review history of their article (what does this mean?). If published, this will include your full peer review and any attached files.

Reviewer #1: No

Reviewer #2: No

---

## [Author Response · Author response to Decision Letter 0]

15 Dec 2021

Dear Mr Najafi,

Thank you for giving us the opportunity to revise and resubmit our manuscript untitled “Comparison of compensatory shoulder movements, functionality and satisfaction in transradial amputees fitted with two prosthetic myoelectric hooks”. We have submitted our revised manuscript in order to make it conform to the additional requirements stipulated in your letter of October 10, 2021. As per your request, we submit this review by the agreed deadline, December 4, 2021. We appreciate the time and detail provided by each reviewer and by you and we have incorporated the suggested changes into the manuscript to the best of our ability. The manuscript has certainly benefited from these insightful revision suggestions. We look forward to working with you and the reviewers to move this manuscript closer to publication in Plos One. 

We have responded specifically to each suggestion below, starting with yours.

Editor’s suggestions:

1- Please ensure that your manuscript meets PLOS ONE's style requirements, including those for file naming. 

We have renamed figures at Fig1.tif.

We have placed the asterisk on the correct corresponding author.

2- In your Methods section, please provide additional information about the participant recruitment method and the demographic details of your participants. Please ensure you have provided sufficient details to replicate the analyses such as a description of any exclusion criteria that were applied to participant recruitment.

In [Methods/Participants] we have provided more details regarding the criteria for recruiting patients.

3- Please ensure you have clearly identified primary, secondary, and any exploratory outcomes.

In our opinion, the different types of results have been clearly specified.

4- Please ensure your patient flow diagram is included as Figure 1 in the main manuscript. We have changed Figure 1 so that it integrates the patients flow diagram.

a. FZB(EO)19Febr21: Registration done retrospectively (after enrollment of participants)

b. FZB(EO)19Febr21: borderline ethics statement (IRB not named)

a. Thank you for submitting your clinical trial to PLOS ONE and for providing the name of the registry and the registration number. The information in the registry entry suggests that your trial was registered after patient recruitment began. PLOS ONE strongly encourages authors to register all trials before recruiting the first participant in a study.

i) your reasons for your delay in registering this study (after enrolment of participants started);

Registration on clinical trial is not a compulsory procedure. In France, the monitoring of studies is well supervised (CPP, CNIL, ANSM). As this study included few patients, it did not seem necessary to us to make this recording. We were not aware at the time of the importance of this clinical trial recording process.

ii) confirmation that all related trials are registered by stating: “The authors confirm that all ongoing and related trials for this drug/intervention are registered”.

We declared that.

b. Thank you for including your ethics statement: "The study protocol was registered with the French National Agency for Medicines and Health Products (ANSM) and approved by an independent ethics committee (CPP Est-III) on August 29, 2016 (N°16.07.02). Data collection complied with the reference methodology for interventional trials (MR-001) issued by the French Data Protection Agency (CNIL).

The subjects were recruited during prosthetic medical appointments usually carried out at the rehabilitation center. During the inclusion period, all persons who met the inclusion criteria were invited to participate in the study. They received all the required oral and written information. All the subjects included in the trial signed the informed consent forms". 

We regularized that by writing the name of the committee in full.

5- We note that you have stated that you will provide repository information for your data at acceptance. Should your manuscript be accepted for publication, we will hold it until you provide the relevant accession numbers or DOIs necessary to access your data. If you wish to make changes to your Data Availability statement, please describe these changes in your cover letter and we will update your Data Availability statement to reflect the information you provide.

We will provide you with access to all data only after acceptance of the manuscript for publication.

6- Please upload a new copy of Figure 3 as the detail is not clear. Please follow the link for more information.

The quality of Figure 3 has been improved.

7- Please include captions for your Supporting Information files at the end of your manuscript, and update any in-text citations to match accordingly. Please see our Supporting Information guidelines for more information.

We edited these captions after [Conclusion].

Reviewer #1:

Major revision: Testing the primary objectives with an all-inclusive linear mixed model would be superior to the analysis approach which has been taken. Mixed models offer the ability to test for differences in the effect of Greifer, Axon-Hook, and non-affected side. T-tests should not be needed. Additionally, specify the underlying covariance structure used in the mixed models and the criteria for selecting it.

We changed the statistical treatment as you advised it. We used a mixed design analysis of variance for repeated measures, with a “Group” factor in order to test if the order of the use of the prosthesis had an impact, and a “Hand” factor as within-subject factor. We have inserted the sentences below in the relevant parts and modified the figures accordingly. 

Minor revisions: 

1- Abstract: We replaced the rho symbol with r which represents the estimate of rho. We also made the changes in the manuscript.

2- Sample size justification:

a. Lines 137-8: “average” was replaced with the standard statistical terminology “mean”.

b. We specified that 25 represent the standard deviation.

The statistical method initially used to determine the number of participants was based on the hypothesis of a comparison of means with a student test for a power of 90% with an alpha threshold of 0.05. Since we have changed the statistical design as you advised, we have chosen to remove this part of the manuscript

3- Line 238-9: The “customary descriptive statistics” were mean and standard deviation.

4- Line 239: Indeed we made a typographical error. We rewrote: “Quantitative variable was described by numbers. Qualitative variables were described by percentages and frequencies”. 

5- Line 247: We replaced the phrase “The alpha-risk was set at 5%” with “P-values less than 0.05 were considered statistically significant.”

6- Throughout the manuscript "average" was replaced by "mean".

7- The p-value associated with a correlation is a test of the null hypothesis: correlation equal to zero; however, the absolute magnitude of the coefficient indicates the strength of the linear relationship between two variables. In general, the strength or correlation coefficient is the more important statistic to focus on.

Below is a table for interpreting correlation coefficients:

Coefficient (absolute value) Interpretation

0.90 - 1.0 Very Strong

0.70 - 0.89 Strong

0.40 - 0.69 Moderate

0.10 - 0.39 Weak

less than 0.10 Negligible correlation

We rewrote the paragraph concerning correlation between shoulder abduction and wrist settings (p.317 à 320) by interpreting correlation coefficients:

“When assessing the effect of wrist settings, a significant strong negative correlation was found between shoulder abduction and flexion with the Axon-Hook (r =-0.86075; p=0.0061). The correlation between shoulder abduction and radial deviation with the Greifer was weak and not significant (r=0.38557; p=0.3455)”.

Reviewer #2:

1) In the paper it is not clear, why the authors have limit themselves just for 2 myoelectric prosthetic devices. Also it seems that conducting this research study limiting just for 2 devices will not provide substantial research outcome.

We have effectively limited this study to two myoelectric hooks. Greifer and Axon Hook were the two most prescribed hooks in France. The Greifer was the most widely used hook because it was the only one covered by the French healthcare reimbursement system. We wanted to compare it with another hook that had different features, which the Axon-Hook responded to. 

2) It is acceptable for having eight participants for this study. But as mentioned in the paper, majority (or some) participants had already being used one of the prosthetic devices that have been considered in this study. So it brings biased experimental output and decrease the quality of research outcome.

Yes indeed it would have been better than no patient has ever used Greifer but:

- We had to include people whose professional or life project would require the use of a myoelectric hook. As stipulated on page 8 of the paragraph [Materials and Methods/Participants] of the manuscript: “their professional activity or their life project justified the use of a myoelectric hook”. So as the Greifer is the most used myoelectric hook in France, some of the participants were already using this device. Axon-Hook was more recent and presented different features than Greifer.

-If it is suggested that experience has a bearing on the compensations, they should therefore have been lower among previous users of Greifer. However, they are more important with the Greifer than with the Axon Hook. This is indeed a methodological bias, but it is not the prior use of Greifer that can explain our results. On the contrary, there is a risk of underestimating the effect.

3) In my point of view, for comparisons like theses, just one experimental test is not enough (i.e. BBT). You can obtain results from multiple test and then thoroughly analyzed. This will bring good research outcome for the audience.

The patients only performed the BBT. Indeed, for technical reasons induced by the handover to the Laboratory (biomechanical model, placement of markers, use of data, etc.), it is difficult to adapt and analyze the results of complex functional motor tests/tasks existing ones. The BBT is the most used one in the literature regarding motion analysis (Kontson and al. 2017(a), 2017(b), 2020); its cyclical nature makes it ideal for motion capture. BBT is a validated timed measure of upper limb performance. Performed in a seated position, it lends itself very well to the evaluation of shoulder abduction.

4) Lastly, it is good to consider several performance metrics, without being stuck into one performance metric (i.e. number of blocks moved within given time) or performance metric within the same aspect (i.e. performance at shoulder movements)

Thank you! Precisely, the Box and Blocks test is the one that best met our objective of simultaneously evaluating these two types of performance metrics.

Constance Billon

---

## [Decision Letter · Decision Letter 1]

4 May 2022

PONE-D-21-05606R1Comparison of compensatory shoulder movements, functionality and satisfaction in transradial amputees fitted with two prosthetic myoelectric hooksPLOS ONE

Dear Dr. BILLON-GRUMILLIER,

Thank you for submitting your manuscript to PLOS ONE. After careful consideration, we feel that it has merit but does not fully meet PLOS ONE’s publication criteria as it currently stands. Therefore, we invite you to submit a revised version of the manuscript that addresses the points raised during the review process.

 Please submit your revised manuscript by Jun 18 2022 11:59PM. If you will need more time than this to complete your revisions, please reply to this message or contact the journal office at plosone@plos.org. Please include the following items when submitting your revised manuscript:A rebuttal letter that responds to each point raised by the academic editor and reviewer(s). You should upload this letter as a separate file labeled 'Response to Reviewers'.A marked-up copy of your manuscript that highlights changes made to the original version. You should upload this as a separate file labeled 'Revised Manuscript with Track Changes'.An unmarked version of your revised paper without tracked changes. You should upload this as a separate file labeled 'Manuscript'.If applicable, we recommend that you deposit your laboratory protocols in protocols.io to enhance the reproducibility of your results. Protocols.io assigns your protocol its own identifier (DOI) so that it can be cited independently in the future. For instructions see: https://journals.plos.org/plosone/s/submission-guidelines#loc-laboratory-protocols. Additionally, PLOS ONE offers an option for publishing peer-reviewed Lab Protocol articles, which describe protocols hosted on protocols.io. Read more information on sharing protocols at https://plos.org/protocols?utm_medium=editorial-emailutm_source=authorlettersutm_campaign=protocols.

We look forward to receiving your revised manuscript.

Kind regards,

Bijan Najafi

Academic Editor

PLOS ONE

Journal Requirements:

Additional Editor Comments (if provided):

I apologize for the delay in the review process of your manuscript. Despite the follow-up with the initial reviewers, except one, the rest they didn't return their comments. To avoid further delay I reviewed your revision and response letter myself. I believe your revision is response to initial critiques. The reviewer 1 has some remaining but valid concerns. These concerns are minors but need to be addressed before I could recommend the acceptance of the manuscript.

Reviewers' comments:

Reviewer's Responses to Questions

**Comments to the Author**

1. If the authors have adequately addressed your comments raised in a previous round of review and you feel that this manuscript is now acceptable for publication, you may indicate that here to bypass the “Comments to the Author” section, enter your conflict of interest statement in the “Confidential to Editor” section, and submit your "Accept" recommendation.

Reviewer #1: (No Response)

2. Is the manuscript technically sound, and do the data support the conclusions?

Reviewer #1: Yes

3. Has the statistical analysis been performed appropriately and rigorously? 

Reviewer #1: Yes

4. Have the authors made all data underlying the findings in their manuscript fully available?

Reviewer #1: No

5. Is the manuscript presented in an intelligible fashion and written in standard English?

Reviewer #1: Yes

6. Review Comments to the Author

Reviewer #1: Minor revisions:

1- Express the correlation coefficients and large p-values to two significant digits.

2- Line 250: Remove the redundant statement , “The threshold for statistical significance (error type I rate) was set to α=0.05.”

3- Line 244: Specify the underlying covariance structure used in the mixed models and the criteria for selecting it.

4- Since no statistical power calculation has been provided, indicate if the study was a pilot or exploratory trial. In brief, provide the rationale for why no formal statistical power justification is needed.

5- If the interaction effect is significant, provide an interpretation of the results, but do not test main effects because the tests for main effects are uninteresting in light of significant interactions. If interaction effects are non-significant, drop the interaction effects from the model and test the main effects. Determining which results to present when testing interactions is often a multi-step process.

6- Lines 323 and 326: Provide more precise p-values, instead of p0.05.

7- In the figures, replace "average" with "mean."

7. PLOS authors have the option to publish the peer review history of their article (what does this mean?). If published, this will include your full peer review and any attached files.

Reviewer #1: No

---

## [Author Response · Author response to Decision Letter 1]

14 Jun 2022

Editor’s suggestion: Please review your reference list to ensure that it is complete and correct. If you have cited papers that have been retracted, please include the rationale for doing so in the manuscript text, or remove these references and replace them with relevant current references. Any changes to the reference list should be mentioned in the rebuttal letter that accompanies your revised manuscript. If you need to cite a retracted article, indicate the article’s retracted status in the References list and also include a citation and full reference for the retraction notice.

We have checked our list of references and related citations, everything is correct.

Reviewer #1:Minor revisions: 

1- Express the correlation coefficients and large p-values to two significant digits: In the abstract and in all the manuscript, we expressed the correlation coefficients and p-values to two significant digits.

2- Line 250: Remove the redundant statement, “The threshold for statistical significance (error type I rate) was set to α = 0.05”. 

We removed the redundancy in brackets: (error type I rate). 

1

3- Line 244: Specify the underlying covariance structure used in the mixed models and the criteria for selecting it.

We had freed ourselves from the "age" and "time to amputation" covariates by ensuring that the patients had no shoulder pathologies. We had already ruled out the

 "Greifer user" covariate. We don’t see other elements that could be covariance factors.

4- Since no statistical power calculation has been provided, indicate if the study was a pilot or exploratory trial. In brief, provide the rationale for why no formal statistical power justification is needed. 

We added in the paragraph 'study design': Based on observations made during recordings in routine clinical practice, the difference between the two mean shoulder abduction was estimated at 25°. To be able to detect such difference with a power of 90 using a threshold for statistical significance of 0.05, we needed to include 8 patients. 

5- If the interaction effect is significant, provide an interpretation of the results, but do not test main effects because the tests for main effects are uninteresting in light of significant interactions. If interaction effects are non-significant, drop the interaction effects from the model and test the main effects. Determining which results to present when testing interactions is often a multi-step process.

Thank you for this comment which has helped us to improve the quality of our statistical analysis. As you requested, we excluded the interaction effect from the analysis when it was not significant. We have specified this in the text as follows: "When the interaction effect was not significant, it was dropped from the analysis and only the main effects were tested."

The result section has been updated to match the new analysis. The discussion has been adapted.

6- Lines 323 and 326: Provide more precise p-values, instead of p0.05. 

We replaced the first p0.05 with p=0.017 and the second with p=0.014.

7- In the figures, replace "average" with "mean."

In Figures 5 et 7 "average" was replaced by "mean".

---

## [Editor Report · Decision Letter 2]

28 Jul 2022

Comparison of compensatory shoulder movements, functionality and satisfaction in transradial amputees fitted with two prosthetic myoelectric hooks

PONE-D-21-05606R2

Dear Dr. BILLON-GRUMILLIER,

We’re pleased to inform you that your manuscript has been judged scientifically suitable for publication and will be formally accepted for publication once it meets all outstanding technical requirements.

Kind regards,

Bijan Najafi

Academic Editor

PLOS ONE

Additional Editor Comments (optional):

Thanks for addressing the remaining critiques. After reviewing your revision and response letter I believe your revision is responsive to all initial critiques and the current revision has significant scientific merit. Thus I recommend acceptance of your revision! Congratulations!
---

## [Editor Report · Acceptance letter]

24 Aug 2022

PONE-D-21-05606R2 

Comparison of compensatory shoulder movements, functionality and satisfaction in transradial amputees fitted with two prosthetic myoelectric hooks 

Dear Dr. Billon-Grumillier:

I'm pleased to inform you that your manuscript has been deemed suitable for publication in PLOS ONE. Congratulations! Your manuscript is now with our production department. 

Kind regards, 

on behalf of

Dr. Bijan Najafi 

Academic Editor

PLOS ONE